# Policy Gradient for items Recommendation on Virtual Taobao

**Wei FANG**
Department of Information Engineering
The Chinese University of Hong Kong
Shatin, Hong Kong
1155145703@link.cuhk.edu.hk

**Fanyuan Zeng**
Department of Information Engineering
The Chinese University of Hong Kong
Shatin, Hong Kong
1155145759@link.cuhk.edu.hk

## Abstract

Recent years have witnessed digital content appear with plenty of forms (including online courses, online shopping and e-news) in daily life of people, which has provided with opportunities as well as challenges for systems to provide users with personalized services and information. The goal of our project is to design a recommender algorithm that can return a good list such that the consumers might have high chance of clicking the items on a simulated environment named Virtual Taobao, a simulator trained from the real-data from Taobao. Firstly, We tried some state-of-art deep-reinforcement algorithms, such as deep deterministic policy gradient (DDPG) method and Twin Delayed DDPG (TD3), what's more, we also used the Proximal Policy Optimisation (PPO) algorithm and tried to improve the PPO algorithm with the features of the attributes of the consumers.Video link:https://drive.google.com/file/d/1WVoAjKcJ-4t5o6n$_U$5KaoDn7BymksYqr/view?usp = sharing

## 1 Introduction

With the development of Information technology, more and more companies pay attention to the recommender system. Recommender systems are widely used in e-commerce websites to recommend products based on the preferences and tastes of online customers to improve their customer's shopping experience. Compared with other technologies, the use of reinforcement learning in the recommender system can better consider the long-term benefits of users, so as to maintain the long-term satisfaction and activity of users in the platform. In this paper, we try to investigate the power of reinforcement learning in this area. we plan to develop a simple recommender algorithm based on a simulated environment. The goal of our algorithm is to increase the mean reward and customer click-through rate (CTR) of the items we recommend.

### 1.1 Related Work

In 2016, the emergence of YoutubeDNN[1] and WDL[2] led the industry trend of recommendation systems and CTR estimation so far, setting off a wave of large-scale and efficient upgrade of deep learning models for recall layer and ranking layer algorithms. In 2019, a Google team achieved the largest reward growth for a single YouTube project in the past two years. The modeling idea is similar to RNN recall, given the user's behavior history, predict the user's next click on the item based on the Off-Policy training strategy[3] . Another Google team's paper also claims that they have achieved good results on the Youtube online recommendation system. The main contribution is to propose a Q-Learning algorithm called SLATEQ, which optimizes the long-term value LTV (Long-term Value) of multiple items displayed to users at the same time in the recommendation system[4] . Also in 2019, a team from Shanghai Jiao Tong University proposed a Tree-structured Policy Gradient Recommendation (TPGR) framework[5] , where a balanced hierarchical clustering

tree is built over the items and picking an item is formulated as seeking a path from the root to a certain leaf of the tree.

## 2  RL Framework

### 2.1  Environment

Applying reinforcement learning in physical-world tasks is extremely challenging. However, reinforcement learning requires a large number of training samples. For example, AlphaGoZero[6] conducted 490 million rounds of simulated Go training, and the reinforcement learning of Atari game ran at high speed on the computer for more than 50 hours[7] . Luckily, we found a virtual environment tool (Virtual Taobao) [8] for reinforcement learning training. The VirtualTaobao simulators trained from the real-data of Taobao, one of the largest online retail platforms. In Taobao, when a customer entered some query, the recommender system returns a list of items according to the query and the customer profile. The system is expected to return a good list such that customers will have high chances of clicking the items. Using virtual simulator, we can access a "live" environment just like the real Taobao environment. Virtual customers will be generated once at a time, the virtual customer starts a query, and the recommendation system needs to return a list of items. The virtual customer will decide if he or she would like to click the items in the list which is similar to a real customer. We can access the environment through OpeanAI gym conveniently, We trained our model on Virtual Taobao and proved our model works.

### 2.2  MDP Modeling

We are now ready to formulate the problem as a Markov Decision Process (MDP) by introducing the agent, states, actions, and the reward.

- Agent: our recommender system

- State: we treat the customers as the states. In the VirtualTaobao environment, A customer is associated with 11 static attributes that has been one-hot encoded in to 88 binary dimensions, and 3-dimensional dynamic attributes. Here, static/dynmaic means whether the features will change during an interactive process. The attributes information about involve customer age, customer gender, customer browsing history, etc.

- Action: the system will select 10 items and push them to the customer. Then, the customer will choose to click on some items or leave the platform (results in the end of the session). The action model of the customers has been trained by the developers of the VirtualTaobao with the data of many real customers.

- Reward: every click on the items will receive +1 reward, and the customer may click all ten items on the page, in addition, they might just end the session without any click.

## 3  Methods and Algorithms

### 3.1  DDPG

Deterministic Policy Gradient (DDPG) [9] is an algorithm which concurrently learns a Q-function and a policy. It uses off-policy data and the Bellman equation to learn the Q-function, and uses the Q-function to learn the policy. This approach is closely connected to Q-learning, and is motivated the same way: if you know the optimal action-value function $Q^*(s, a)$.

In our actor-critic neural networks, we use two-layer MLP with each layer containing 128 hidden layers to construct our networks. In addition, we add layer normalization after each layers, the details of the algorithm is shown as below.

**Algorithm 1** Deep Deterministic Policy Gradient

1: Input: initial policy parameters $\theta$, Q-function parameters $\phi$, empty replay buffer $\mathcal{D}$
2: Set target parameters equal to main parameters $\theta_{\text{targ}} \leftarrow \theta$, $\phi_{\text{targ}} \leftarrow \phi$
3: **repeat**
4:   Observe state $s$ and select action $a = \text{clip}(\mu_\theta(s) + \epsilon, a_{Low}, a_{High})$, where $\epsilon \sim \mathcal{N}$
5:   Execute $a$ in the environment
6:   Observe next state $s'$, reward $r$, and done signal $d$ to indicate whether $s'$ is terminal
7:   Store $(s, a, r, s', d)$ in replay buffer $\mathcal{D}$
8:   If $s'$ is terminal, reset environment state.
9:   **if** it's time to update **then**
10:     **for** however many updates **do**
11:       Randomly sample a batch of transitions, $B = \{(s, a, r, s', d)\}$ from $\mathcal{D}$
12:       Compute targets

$$y(r, s', d) = r + \gamma(1 - d)Q_{\phi_{\text{targ}}}(s', \mu_{\theta_{\text{targ}}}(s'))$$

13:       Update Q-function by one step of gradient descent using

$$\nabla_\phi \frac{1}{|B|} \sum_{(s,a,r,s',d) \in B} (Q_\phi(s, a) - y(r, s', d))^2$$

14:       Update policy by one step of gradient ascent using

$$\nabla_\theta \frac{1}{|B|} \sum_{s \in B} Q_\phi(s, \mu_\theta(s))$$

15:       Update target networks with

$$\phi_{\text{targ}} \leftarrow \rho\phi_{\text{targ}} + (1 - \rho)\phi$$
$$\theta_{\text{targ}} \leftarrow \rho\theta_{\text{targ}} + (1 - \rho)\theta$$

16:     **end for**
17:   **end if**
18: **until** convergence

### 3.2 PPO

PPO-clip [10] updates policies via

$$\theta_{k+1} = \arg\max_\theta \mathop{\mathrm{E}}_{s,a \sim \pi_{\theta_k}} [L(s, a, \theta_k, \theta)],$$

typically taking multiple steps of (usually minibatch) SGD to maximize the objective. Here L is given by

$$L(s, a, \theta_k, \theta) = \min\left(\frac{\pi_\theta(a|s)}{\pi_{\theta_k}(a|s)} A^{\pi_{\theta_k}}(s, a), \ \text{clip}\left(\frac{\pi_\theta(a|s)}{\pi_{\theta_k}(a|s)}, 1 - \epsilon, 1 + \epsilon\right) A^{\pi_{\theta_k}}(s, a)\right),$$

in which $\epsilon$ is a small hyperparameter which roughly says how far away the new policy is allowed to go from the old.

This is a pretty complex expression, and it's hard to tell at first glance what it's doing, or how it helps keep the new policy close to the old policy. As it turns out, there's a considerably simplified version of this objective which is a bit easier to grapple with (and is also the version we implement in our code):

$$L(s, a, \theta_k, \theta) = \min\left(\frac{\pi_\theta(a|s)}{\pi_{\theta_k}(a|s)} A^{\pi_{\theta_k}}(s, a), \ g(\epsilon, A^{\pi_{\theta_k}}(s, a))\right),$$

where

$$g(\epsilon, A) = \begin{cases} (1 + \epsilon)A & A \geq 0 \\ (1 - \epsilon)A & A < 0. \end{cases}$$

$$\theta_{k+1} = \arg\max_{\theta} \frac{1}{|\mathcal{D}_k|T} \sum_{\tau \in \mathcal{D}_k} \sum_{t=0}^{T} \min\left( \frac{\pi_\theta(a_t|s_t)}{\pi_{\theta_k}(a_t|s_t)} A^{\pi_{\theta_k}}(s_t, a_t), \; g(\epsilon, A^{\pi_{\theta_k}}(s_t, a_t)) \right),$$

 typically via stochastic gradient ascent with Adam.
7: Fit value function by regression on mean-squared error:

$$\phi_{k+1} = \arg\min_{\phi} \frac{1}{|\mathcal{D}_k|T} \sum_{\tau \in \mathcal{D}_k} \sum_{t=0}^{T} \left( V_\phi(s_t) - \hat{R}_t \right)^2,$$

 typically via some gradient descent algorithm.
8: **end for**

### 3.3   Improved PPO

As mentioned before, the states (customers) is associated with 88 binary dimensions attributes and 3-dimensional dynamic attributes. So we decided to train the two features respectively. In the actor neural network, we use the three-layer MLP with each layer containing 128 hidden neurons to tain the two features respectively (The same as our PPO-Clip method). Then we use a full connected layer to combine the outputs of the two MLP. What's more, we think our policy should emphasize on the update of the 3-dimensions dynamic attributes, so we set the update frequency of the 3-dimensions MLP is one hundred times that of the other one. Other parts are just the same as the PPO method in the part 3.2.

## 4   Experiments

So far we have finished three experiments on the Google Colab platform, including DDPG algorithm, PPO algorithm and our improved PPO algorithm, Here are the results.

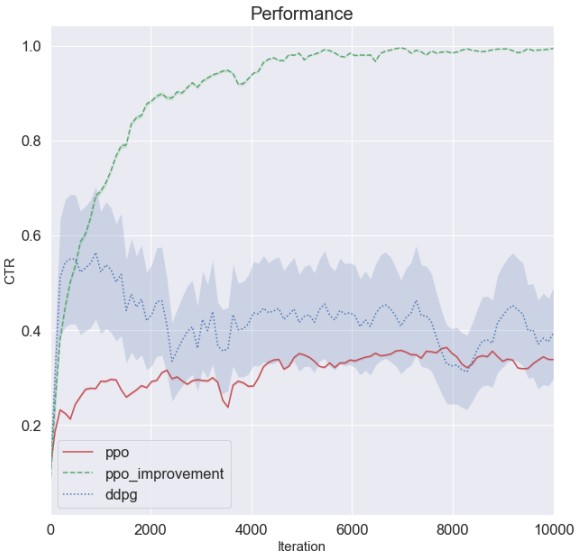

Figure 1: CTR

The figure shows the mean CTR of 50 states generated randomly, We can learn from the the figure that the CTR of DDPG increase rapidly from 0 to about 0.5 between 0 and 200 iterations, however it drops dramatically to around 0.42 after 2000 iterations, after that it fluctuates around 0.42. PPO behaves more stable, it doesn't drop during the training, but it increases from 0 to 0.37 but never larger than 0.4. The performance of our improved PPO method is much better the the DDPG and PPO method, it increase form 0 to 1 stably and it reach the maximum with only 5000 iterations and nearly get 1 CTR for one page items. It means customer will choose to click all ten items on one page we recommend.

## 5 Conclusions

The DDPG and PPO method don't have good performances. Our improved PPO method behaves much better than we expected. It tells that our improvement based on the features of the attributes of the customers are successful, but this may leads to the dangerous area of the overfitting. What's more, our experiments are based on one page items, we may do more investigations about the customers that may choose to view more pages which is more closer to the reality in the future.

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
