# OpenReview forum: "Policy Gradient for items Recommendation on Virtual Taobao"
_CUHK.edu.hk/2021/Course/IERG5350_

### Official Review · AnonReviewer2 · 2020-12-18
**this report aims to leverage improved policy gradient method  to improve the performance of items recommandation on virtual taobao**

**Rating:** 6
**Confidence:** 4

**Review:**

This report aims to leverage improved policy gradient method  to improve the performance of items recommandation on virtual taobao. Based on PPO, they design a network and corresponding training strategy to improve the performance. Overall, their improvement is effective in the experiment results, but some parts of this report are not clear enough and experimental results are incomplete. Detailed comments are in the following.

1, The topic they choose is valuable and interesting. The recommandation system is widely used in electronic commerce platforms.

2, The report is well organized and easy to follow.

3, According to the characteristics of the state (dynamic part and static part, dynamic part seems to be more important), they design a new network and corresponding training strategy. They compare their method with DDPG, PPO. From the experiment results, their method outperforms these two methods with a large margin.

4, This report doesn't introduce the background of their problem in detail. For example, they don't give a detailed introduction to attributes included in state and why dynamic attributes are more important than static attributes. These background are very important to help readers apprecaite the proposed work.

5, This report shows the algorithm of DDPG and PPO but doesn't show the algorithm of their own method. This may cause confusion of readers. For example, they set the update frequency of the 3-dimensions MLP is one hundred times that of the other one. Does this mean that in one iteration, they train the 3-dimensions MLP 100 times and the other one once? Then how many times they train MLP in one iteration for PPO and DDPG?

6, This report doesn't show their experiment details. For example, they don't state how many times they run when testing. It seems they run DDPG multi times and calculate mean value and  standard deviation but run PPO and PPO_improved only once. If so, it's not a fair way to compare these algorithms.

7, The report and the video they present are both short in length and feel like not well prepared. The report is less than 5 pages and the video is 3 minutes and 33 seconds.

---

### Official Review · AnonReviewer3 · 2020-12-20
**Improved PPO based on state features but lacks analysis and experiment details**

**Rating:** 6
**Confidence:** 4

**Review:**

General:
The paper explores the application of DDPG, PPO on Taobao recommendation system, where they tried to maximize customer click-through rate (CTR). Based on the state features of the attributes of customers provided by the virtual Taobao, they propose improved PPO to train different features at different rates

Evaluation of the quality:
The main contribution of this work is to customize the PPO based on the state features of the environment. The experiment results seems solid to show the improvement, but lacks comparison analysis of performance.

Clarity:
	1. Clear definition of the recommendation system problem.
	2. Not much introduction to the virtual Taobao and the different features of the customers

Originality:
As there are no related work based on the virtual Taobao listed in the paper, I can hardly tell the originality.

Significance:
I didn't see any significance of training such recommendation system in a virtual Taobao. But the exploration of different networks on different features is interesting.

Pros:
	1. The experiment results are promising.

Cons:
	1. Lacks full details of the environment.
	2. Lacks details about the experiment setups and comparison analysis performance.